# Risk Factors for Mortality in Colombian Patients with Candidemia

**DOI:** 10.3390/jof7060442

**Published:** 2021-05-31

**Authors:** Jorge Alberto Cortés, Anita María Montañez, Ana María Carreño-Gutiérrez, Patricia Reyes, Carlos Hernando Gómez, Angela Pescador, Beatriz Ariza, Fernando Rosso

**Affiliations:** 1Department of Internal Medicine, Facultad de Medicina, Hospital Universitario Nacional, Universidad Nacional de Colombia, Bogotá 111321, Colombia; ammontaneza@unal.edu.co (A.M.M.); amcarrenog@unal.edu.co (A.M.C.-G.); 2Department of Infectious Diseases, Clínica Universitaria Colombia, Bogotá 111321, Colombia; patriciareyes@colsanitas.com; 3Hospital Militar Central, Bogotá 111321, Colombia; carlosgomez1074@gmail.com (C.H.G.); luzangelapv@hotmail.com (A.P.); 4Hospital Universitario San Ignacio, Bogotá 111321, Colombia; deariza@husi.org.co; 5Fundación Valle de Lili, Cali 760026, Colombia; frosso07@gmail.com

**Keywords:** candidemia, mycoses, *Candida*, *Candida albicans*, intensive care units, mortality, survival analysis

## Abstract

The aim of the study was to describe the microbiology and susceptibility profile of candidemia and to identify the risk factors associated with mortality in Colombia. A cohort of patients was followed for 30 days during 2008 to 2010. Microbiological identification and susceptibility assessments were performed in a reference centre. Demographic, clinical and treatment variables were evaluated for their associations with mortality. A parametric survival regression analysis was used to identify the risk factors associated with mortality. A total of 109 patients with candidemia in four hospitals in Colombia were identified, with a median age of 30 years old. *C. parapsilosis* was the most frequently identified microorganism (38.5%); the susceptibility of all isolates was high to fluconazole and anidulafungin, except for *C. glabrata* isolates. The overall mortality was 35.7%, and the risk factors associated with mortality included lack of antifungal treatment (HR 5.5, 95% CI 3.6–11.4), cancer (HR 3.9, 95% CI 2.3–8.0), diabetes (HR 2.5, 95% CI 1.03–6.4), and age (HR 1.13 per every 10 years, 95% CI 1.02–1.24). Catheter removal was associated with a low mortality rate (HR 0.06, 95% CI 0.00–0.49). Prompt antifungal treatment, better glycemic control and catheter removal should be prioritized in the management of candidemia.

## 1. Introduction

Candidemia is the fourth leading cause of infection in the bloodstream in the United States and the sixth cause in Latin America [1,2], as well as the most common fungal infection in hospitalized patients, especially in critically ill patients, independent of immunological status. The mortality attributable to candidemia remains high, at approximately 20 to 50% [3,4,5], and therefore, the relevance of the knowledge of the epidemiology, risk factors and susceptibility to antifungal drugs of the different species are important for clinicians who face patients with this infection.

There are important differences in the hospital-acquired infections between Latin American countries and other continents. A higher number of nosocomial infections have been described in the region [6], and studies have shown a relatively high incidence of candidemia with a high mortality [7]. Between 2008 and 2010, a surveillance study of candidemia was performed in seven Latin American countries and showed a high incidence of candidemia and slightly increased numbers in Colombia in comparison to those of the region region (1.96 cases per 1000 admissions vs. 1.18 cases per 1000 admissions) [5,8].

The present study presents the results of a laboratory-based surveillance of candidemia in Colombia as part of the Latin American effort to improve the knowledge of Candida epidemiology in the region.

## 2. Materials and Methods

### 2.1. Setting

This is a prospective laboratory-based surveillance study conducted between November 2008 and October 2010 (24 months) in 4 tertiary care hospitals in Colombia (a total of approximately 1300 beds). They were all general hospitals, and two were in the public sector. All hospitals treated patients of all ages and had intensive care units (ICUs, 160 beds) and departments of internal medicine and surgery; three centres had a haematology ward, two had a solid organ transplantation ward, and one had a haemopoietic cell transplantation ward. Patients from one hospital (Fundación Valle de Llili-FVL) were mostly paediatric and included neonates. Patients from other Hospital (Hospital Universitario San Ignacio-HUS) were adults. One general hospital was a referral center for the military health services in the country (Hospital Militar Central-HMC) and the other was a reference center for general population (Hospital Universitario San Ignacio -HUSI). There was no antifungal protocol in use in any the hospitals and no prophylaxis strategy for the prevention of such infections in any service of the institutions. All hospitals had automated blood culture systems (Bactec, Becton Dickinson, NJ, USA). The case report form contained detailed information about demographics, underlying conditions, coexisting exposures, receipt of antifungal agents and the outcome. All clinical information was sent using a web-based system (SPSS, Inc., New York, NY, USA).

All adult and paediatric patients with a microbiological diagnosis of candidemia were eligible for inclusion in the study and were followed for 30 days. There were no exclusion criteria.

### 2.2. Definitions

One episode of candidemia was defined by the isolation of a *Candida* species from one or more blood cultures in a patient with clinical signs of infection. If more than one blood culture was positive, a new episode was defined if more than 30 days had elapsed since the first positive blood culture (incident candidemia). Elderly patients were defined as those older than 60 years. Cancer was defined as the presence of a diagnosis of active cancer or current treatment for identified cancer. Shock was defined as the presence of hypotension on the day that candidemia was identified (systolic arterial pressure of less than 90 mmHg). Delayed treatment was considered when the antifungal was started more than 48 h of the day the blood sample was taken. 

### 2.3. Microbiological Procedures

All isolates were identified at the species level in the local laboratory and sent to a reference laboratory within the country (Laboratorio de Micología, Hospital Universitario San Ignacio) to confirm the species, as well as to perform antifungal susceptibility tests. Isolates were identified according to their microscopic morphology on cornmeal-Tween 80 agar, complemented by biochemical tests using the ID 32C system (BioMérieux, Marcy l’Etoile, France) or the Yeast ID panel (MicroScan, Beckman Coulter, Brea, CA, USA). Antifungal susceptibility tests were performed using a broth microdilution assay following the methods recommended by the Clinical and Laboratory Standards Institute (CLSI) [9]. The following antifungal drugs were tested: fluconazole (Pfizer Incorporated, New York, NY, USA), amphotericin B (Sigma Chemical Corporation, St. Louis, MO, USA), voriconazole, and anidulafungin (Pfizer Incorporated, New York, NY, USA). The assays were incubated at 35 °C for 24 h. The minimum inhibitory concentration (MIC) breakpoints used were those reported in the literature [10]. For amphotericin B, isolates that responded to MICs ≤ 1 mg/mL were considered susceptible and those that responded to MICs ≥ 2 mg/mL were considered resistant, although no official breakpoints have been established. 

### 2.4. Ethical Approval

The protocol was approved by the ethical and research review board of each of the following hospitals: Comité de Investigaciones y Ética Institucional (CIEI) de la Facultad De Medicina de la Pontificia Universidad Javeriana and from Hospital Universitario de San Ignacio; Comité de Investigación, Fundación Valle del Lili; Comité Independiente de Ética en Investigación, Hospital Militar Central; and Comité Ético Científico de la Empresa Social del Estado Hospital Universitario de la Samaritana. Informed consent was not required by any of the institutional review boards of the participating hospitals since no intervention was performed.

### 2.5. Statistical Analysis

Dichotomous variables were compared using Fisher’s or chi-square tests, as appropriate, and continuous variables were compared using the *t*-test or the Wilcoxon test, as appropriate. All statistical analyses were performed in Stata (v. 15.0, TX, USA). Main outcome was defined as all-cause mortality during the hospital stay or in the following 30 days of the incident candidemia. To identify the independent risk factors associated with mortality, an initial bivariate analysis was performed, and the variables with p values less than 0.2 or clinical significance were considered for inclusion in the survival analysis. The survival functions were estimated with the Kaplan-Meier method for each subgroup. For the multivariable analysis, a parametric survival analysis was performed since the proportional hazards assumption was not met. Different models (Weibull, exponential and lognormal) were fitted using an accelerated failure time to mortality, and they were compared using the Akaike information criteria (AIC) values. Hazard ratios and their 95th percentiles were estimated using the negative exponential ratio of the coefficient to the scale parameter obtained in the best model. Survival calculations were performed using R (v. 3.4.3, survival package).

## 3. Results

During the study, 109 cases of candidemia were identified. The median age of the patients differed among hospitals, as shown in Appendix A. The median age of the whole group was 30 years (range 0–88 years). There were 31 (28.4%) paediatric patients, with 18 (16.5%) patients younger than 1 year old (including 13 neonates), 9 (8.3%) patients between 1 and 12 years old and 4 (3.7%) patients between 13 and 18 years old. Of the 78 adults, 26 (33% of the adults) were older than 60 years. The median age of the newborns was 23 days (IQR 9–28 days).

The main risk factors for candidemia observed in the patients, as well as other demographic data, are presented in Table 1. 69 (63.3%) patients underwent surgery, and 29 (26.6%) of them had at least one other surgical procedure (12–11%- of these second surgeries were abdominal procedures). The immunosuppressed patients included 2 (1.8%) patients with autoimmune diseases, 2 (1.8%) with transplanted patients (one kidney transplant and one liver transplant) and one (0.9%) patient with HIV infection. Of the patients with cancer, 7 (6.4%) were neutropenic at the time of diagnosis. Two (1.8%) patients had burns (burned area between 18 and 24% of the body’s surface area).

The patients had been in the hospital for a median of 19 days (IQR 9–30) before the incident candidemia. At the time of diagnosis, the Apache score was available for 66 adult patients with a median of 13 points (IQR 7–18). A total of 39 (50%) of the adult patients and 21 (67.7%) of the paediatric patients were in the intensive care unit at the time of diagnosis. Previous mechanical ventilation was utilized in 53 (67.9%) adult patients and 12 (38.7%) paediatric patients, and a central catheter was placed in 62 (79.5%) of the adult patients and in 24 (77.4%) of the paediatric patients. Median time of catheter use was 12 days (IQR 7–19 days). The median time after the start of parenteral nutrition and the identification of candidemia was 17 days (IQR 10–24 days). The central line was removed in 12 (15.4%) adult patients and 5 (16.1%) paediatric patients. At the time of diagnosis, 88 (80.7%) patients had fever, 62 (56%) took inotropic medication, and 25 (22.9%) were in shock. All patients had received at least one antibiotic, fifty-one (46.8%) patients had up to 2 antibiotics, and fifty-two (47.7%) patients had received between three and four antibiotics at the time of diagnosis. Thirty-three (30.3%) patients had received a previous antifungal agent.

### 3.1. Microbiological Findings

*C. parapsilosis* was the most frequently identified species (*n* = 42, 38.5%), followed by *C. albicans* (*n* = 40, 36.7%), *C. tropicalis* (*n* = 19, 17.4%), *C. glabrata* (*n* = 5, 4.6%), and two cases of *C. guillermondii* and one of *C. lusitaniae*. No differences were found in the distribution of the *Candida* species according to age or immunosuppression status. There was a trend for *C. tropicalis* to be more frequently found in the adult ICU than in the paediatric ICU (*p* = 0.056). The results of the susceptibility test are presented in Table 2. All of these patients had isolates that were susceptible to amphotericin. Six isolates were found in the susceptible dose-dependent (SDD) category to fluconazole (one isolate of *C. parapsilosis* and the five isolates of C. glabrata), and none were resistant. One isolate was found to be intermediately susceptible to voriconazole (of 101 isolates with a specific breakpoint), and four of the five isolates of *C. glabrata* had high MICs. Regarding anidulafungin, five isolates were found in the intermediate category of those with specific breakpoints (5 of 106, 4.6%) and no resistance. Six (5.5%) patients had bacteraemia at the same time that candidemia was detected (2 patients with *Klebsiella spp*., and one patient each had coagulase negative *Staphylococcus, Enterococcus spp*., and other gram-negative bacteria). Forty-three (39%) patients had a previous episode of bacteraemia. In 17 (15.6%) patients, a second blood culture was positive for the same *Candida* species (15.6%). The median time to the second positive blood culture was 7 days (IQR 5–11 days).

### 3.2. Treatment and Outcome

Treatment with at least one antifungal was started in 96 patients (88.1%). Treatment was started with a median delay of 2 days after the blood sample was taken(IQR 1–4 days). The treatment was started on the same day (of the blood culture) in 16 patients (16.6%), the next day in 11 patients (11.4%), and after 2 days in 69 patients. The median time to starting treatment was longer for the group of patients over 60 years old, in which the treatment was started a median of 3.5 days (IQR 2–7) after the blood culture was taken. Antifungal treatment was started with fluconazole in 58 cases (60%), amphotericin B (deoxycholate) in 20 cases (20.8%) and caspofungin in 8 cases (8.3%). The treatment was changed in 24 patients after a median time of 5.5 days from the start of the first antifungal (IQR 3–10.5 days). Amphotericin B treatment was changed in 11 cases (55% of the patients with that drug); it was changed to fluconazole on 8 occasions and to caspofungin on 2 occasions. Fluconazole was changed to caspofungin in 4 cases and to amphotericin in 3 cases. A third antifungal treatment was administered in 11 cases.

Treatment was administered for a median of 14 days (IQR 9–21 days). Overall, 20 (18.3%) patients died in the first week, 24 (25%) died in the first 15 days, and 39 (35.7%) died in the 30 days of follow-up. Table 3 shows the differences between surviving and non-surviving patients according to the variables with p values less than 0.2 between the two groups. 

### 3.3. Risk Factors for Mortality

Variables that were statistically significant at the bivariant analysis were considered for inclusion in the survival regression model. No differences were found in mortality according to the unit (intensive care unit, etc.) where the patients were at the time of candidemia diagnosis. Figure 1 shows the survival analysis of the patients according to the start of the antifungal treatment. Figure 2 shows the survival analysis of the patients according to the presence of cancer. The identified risk factors for mortality in the final Weibull model included those in Table 4. Risk factors for mortality included lack of treatment, age, cancer and diabetes, while reduced mortality was associated with catheter removal. Delayed treatment and differences in the antifungal treatment were not statistically significant and were not included in the final model.

## 4. Discussion

Our study shows important risk factors for mortality among patients with candidemia in Colombia. As previously reported, a lack of antifungal treatment is associated with increased mortality after adjusting for various risk factors. Although recognized as a serious infection, candidemia might not be treated with antifungal medication, and a high proportion of these patients receive delayed treatment. A previous study in Spain showed that early treatment (with a similar definition as that used in our study) was associated with reduced mortality [11]. Another study in the USA identified a time window of 24 h among patients in shock, as well as adequate source control, as the key factors to diminishing mortality. In Latin America, the mortality is similar to that reported in other countries, and the proportion of patients who received treatment was also similar [5]. In Colombia, a previous study showed that a similar proportion of patients were treated with antifungal medications and had a similar mortality [12], showing that there is a need for prompt recognition and treatment initiation in patients with candidemia, and probably reflecting the need for stewardship programs on antifungals to provide appropriate treatments in those that require it, and to avoid unnecessary use among the groups at risk. A large cohort study in Spain identified different risk factors for mortality according to the time of death [13]. In the first week after microbiological identification, early appropriate antifungal treatment, catheter removal, the Acute Physiology and Chronic Health Evaluation II score and abdominal source were independently associated with early mortality. Late mortality, occurring in the 30 days after identification, was independently associated with age, intubation, renal replacement therapy and the primary source. Our study identified risk factors from both groups. Interestingly, non-survivors had a lower frequency of persistent candidemia, which might be due to the high mortality observed in the first days after the identification, specially in those without treatment.

Unlike previous studies, our study identified that diabetes as an independent risk factor for mortality. A study performed in Italy showed that older patients usually have more comorbidities [14] and that diabetes was more commonly found in these older patients. In another study from Brazil, diabetes was found to be related to an outbreak of fluconazole-resistant *C. parapsilosis* [15]; however, diabetes was not found to lead to an increase in mortality. The reasons why diabetes increases the mortality in our study might include poor glycaemic care and more comorbidities among patients with diabetes (related to poor chronic care), or diabetes might be a confounding factor related to surgical site infections (surgery is a risk factor for candidemia) or to the development of candidemia [16,17]. Diabetes has also been related to infections of *C. albicans* in China [18]. A previous study from Kansas City that included patients with diabetes and candidemia infections showed that mortality was related to the severity of illness, as measured by the Apache score, mechanical ventilation and nosocomial appearance of the infection [19]. In our study, no differences were found in mortality according to the unit where the patient was identified with candidemia, unlike in other studies [20]. This finding might be related to the availability of critical care or differences in protocols and technology used around the world.

In middle-income countries, such as those in Latin America, there has been a progressive change in epidemiology, with an increase in mortalities caused by cardiovascular causes, an increase in access to critical care units and an increase in the incidence of candidemia among such units [21]. There has also been a shift in the proportion of the predominant *Candida* species. Our study is in line with a Latin American surveillance study [5] and with other surveillance studies in Brazil [22] that observed a decreasing number of *C. albicans* isolates and an increase in the isolates of other species, especially *C. parapsilosis* isolates. The number of isolates of *C. glabrata* is not high, although other studies have shown some elevated numbers [23], and in more recent years, outbreaks of *C. auris* have been detected in some areas in the country [24].

Our results also showed that *Candida* spp. had mild resistance to antifungals. An advantage of this study is that susceptibility testing was performed according to a reference methodology [25]. Previous studies in the country have shown high resistance, but they were performed with different methodologies [12,26]. However, resistance might be higher among certain risk groups, such as immunocompromised patients and, perhaps, those treated with rifampin [27]. Recent studies have shown mild resistance to anidulafungin and some cases of high resistance to caspofungin [23], which might be related to the high costs of these antifungals and to the fact that anidulafungin is unavailable from the national formulary. Our study showed that fluconazole was the preferred antifungal treatment at the time the study was performed. No relationship with fluconazole susceptibility and clinical success was found, but this issue remains controversial [28], and the limited number of resistant isolates precludes further investigation. Our study was performed before the recognition of *C. auris* outbreaks in the country [24], which have been mostly limited to hospitals on the Caribbean Coast; the incidence of these outbreaks has not been reported, and most likely, many hospitals are not able to identify these isolates correctly.

This study has some limitations, such as the relatively small number of cases in high complexity hospitals. In addition, the study was developed in four hospitals in two cities, which implies some geographic restriction because of the type and number of institutions included. Previous data from intensive care units in Colombia have suggested a higher incidence than that reported by this study [29].

## Figures and Tables

**Figure 1 jof-07-00442-f001:**
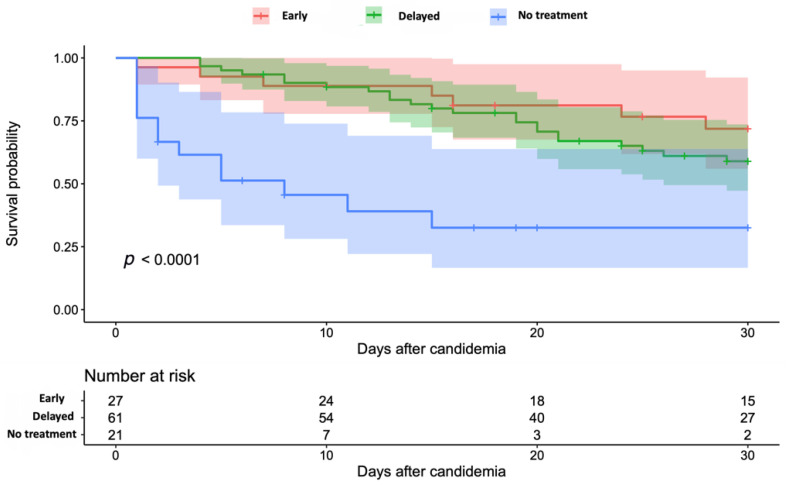
Survival analysis of patients with candidemia according to the time of start of treatment. Note: The coloured shadow shows the 95% confidence interval. Time was counted beginning the day the blood sample was taken.

**Figure 2 jof-07-00442-f002:**
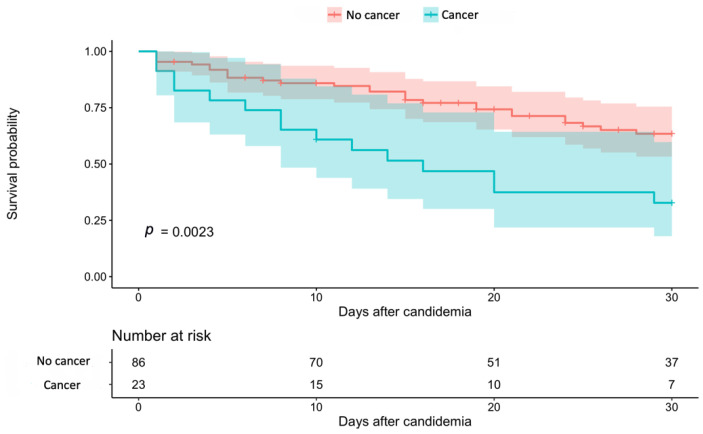
Survival analysis of patients with candidemia according to the presence of cancer. Note: The coloured shadow shows the 95% confidence interval.

**Table 1 jof-07-00442-t001:** Demographic data and identified risk factors among patients with candidemia.

Characteristic or Risk Factor	*N* (%)
Age group	
Neonate (<30 days)	18 (16.5)
Paediatric (30 days to <18 years)	13 (11.2)
Adult (≥18 years to ≤60 years)	52 (47.7)
Elderly (>60 years)	26 (23.9)
Male	74 (67.9)
Central catheter	88 (80.7)
Parenteral nutrition	74 (67.9)
Total parenteral nutrition	65 (59.6)
Mechanical ventilation	65 (59.6)
ICU admission	60 (55.0)
Surgery	69 (63.3)
Abdominal	34 (31.2)
Orthopaedic	10 (9.1)
Other	35 (32.1)
Cancer	24 (22)
Solid	16 (14.7)
Chronic pulmonary diseases	18 (16.5)
Renal failure	16 (14.7)
Acute	11 (10.1)
Cardiac failure	13 (11.9)
Diabetes	13 (11.9)
Chronic liver disease	6 (5.5)
Immunosuppression	5 (4.6)
Previous antibiotic use	109 (100)
Previous antifungal use	33 (30.3)
Neonate and paediatric	11 (35.5)
Adult and elderly	22 (28.2)
Total number of patients	109 (100)

**Table 2 jof-07-00442-t002:** Susceptibility results against selected antifungals.

Species (No. of Isolates)	Antifungal	Range	MIC 50 mcg/mL	MIC 90 mcg/mL	Non-Susceptible (%)
*C. parapsilosis* (*n* = 42)	Amphotericin	0.25–1	0.5	1	0
	Anidulafungin	0.25–2	1	2	0
	Fluconazole	0.125–2	0.25	2	2.4
	Voriconazole	0.03–0.25	0.03	0.03	0
*C. albicans* (*n* = 40)	Amphotericin	0.25–1	1	1	0
	Anidulafungin	0.03–1	0.25	0.25	2.5
	Fluconazole	0.125–4	0.25	0.25	0
	Voriconazole	0.03–0.06	0.03	0.03	0
*C. tropicalis* (*n* = 19)	Amphotericin	0.25–1	0.5	1	0
	Anidulafungin	0.03–0.5	0.125	0.5	0
	Fluconazole	0.125–0.5	0.25	0.5	0
	Voriconazole	0.03–0.06	0.03	0.06	0
*C. glabrata* (*n* = 5)	Amphotericin	0.5–1	1	1	0
	Anidulafungin	0.03–0.25	0.125	0.25	40
	Fluconazole	2–8	4	8	100
	Voriconazole	0.06–0.5	0.125	0.5	N.D.
Other species (*n* = 3)	Amphotericin	0.25–0.5	0.5	0.5	0
	Anidulafungin	0.06–1	1	1	66 *
	Fluconazole	0.125–2	0.125	2	0
	Voriconazole	0.03–0.06	0.03	0.06	N.D

* With C. guillermondii break points. N.D. = Not defined.

**Table 3 jof-07-00442-t003:** Characteristics of patients according to the final outcome (survival).

	Survivors *N* = 66	Non-Survivors *N* = 43	Difference (95% CI)
Age (years) *	29.8	46.0	16.2 (5.8–26.6)
Age > 60 years (%)	15.2	37.2	22.0 (5.2–38.9)
Previous length of stay (days)	19.7	28.4	8.8 (0.3–17.2)
Shock (%)	15.2	34.9	19.7 (3.1–36.4)
Apache (points) **	12.1	15.3	3.2 (−13.2–6.5)
Cancer (%)	12.1	34.9	22.8 (6.5–39.0)
Abdominal surgery (%)	24.2	41.9	17.6 (−0.4–35.6)
Persistent candidemia (%)	19.7	9.3	10.4 (−2.5–23.3)
Catheter removal (%)	24.2	2.3	21.9 (10.6–33.2)
Early treatment (%)	30.3	16.3	14.0 (−1.6–29.7)
No treatment (%)	12.2	30.2	18.0 (2.3–33.9)
Fluconazole as initial treatment(%)	60.6	41.9	18.7 (−0.1–37.6)

* Mean, ** Data available for 90 patients (55 survivors and 35 non-survivors).

**Table 4 jof-07-00442-t004:** Identified risk factors for mortality in patients with candidemia in Colombia.

Risk Factor	Hazard Ratio	95% Confidence Interval
No antifungal treatment	5.52	3.56–11.42
Cancer	3.93	2.34–8.01
Diabetes	2.57	1.03–6.40
Age (per 10 years)	1.13	1.02–1.24
Catheter removal	0.06	0.00–0.49

## Data Availability

De-identified data is available at Dryad Dataset: https://doi.org/10.5061/dryad.gxd2547kq, accesed on 9 April 2021.

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
