# Peer review of "Risk Factors for Mortality in Colombian Patients with Candidemia"

_jof, 2021, doi:10.3390/jof7060442_

Round 1

Reviewer 1 Report

Cortés, et al. describe a sub-investigation performed as part of a larger 7-country Latin American candidemia surveillance study. Specifically, they examine risk factors for mortality among people with candidemia in 4 hospitals. With increasing attention on candidemia in the past few years, it is valuable or would be of interest to readers to describe and share these mortality risk factors. While it’s well-written overall, the paper could be and, in some places, must be edited to correctly interpret the results. Here are some  considerations for the authors:

Abstract

  • L31: The catheter removal CI should be 0.00-0.49 according to table 3.

Introduction

  • L38-39: For a paper about Colombia, it seems odd to have the opening sentence be about the United States. Is there perhaps a more general statement or a Colombia/Latin America/South America statement that could work there instead?

Methods

  • L75: Should this say “…current treatment for identified cancer”? Or is this intending to say current treatment for cancer overlapping with the time of having candidemia (more similar to the shock definition)?
  • Would it be possible to define the main outcome, mortality? It reads like this is describing all-cause mortality, but it would be helpful to clarify for the reader.

Results

  • L126-127: Could you put the number of ‘patients who underwent surgery’ in this sentence? It unclear currently whether you are saying 29 of the 69 patients in table 1 with surgery actually had >=2 surgeries or not and if the 12 abdominal procedures are included in the 34 of the table or not. Perhaps putting the number of surgeries patients had in the table would help?
  • Table 1: Can the ages be added next to the age groups?
  • Table 1: Can the total N be added to the table and clarify how the numbers are being presented? For instance, I initially assumed the table was out of 109 cases, but when you add up the numbers in each age group, it adds up to 153 yet the percentages for each age group appear to be out of 109 (e.g. 26/109 = 23.9%).  Is it that there are 109 unique people involved but a substantial number moved up an age group between different candidemia episodes?
  • Table 1: What is the time period for central line, parenteral nutrition, and surgery?
  • Table 1: There are a substantial number of things that seem like they would fit into table 1 that are mentioned in the text but are not presented in table 1 (e.g. APACHE score, mechanical ventilation, antifungals, etc.). I imagine readers may appreciate those being added to table 1.
  • L153-154: The description of the susceptibility pattern as ‘excellent’ would be more appropriately placed in the discussion.
  • L173: Is it correct that the upper end of the IQR is 70 days? Based on the 30-day candidemia episode criteria and the 30 day follow-up period, I would imagine that 70 days later would be counted towards a different candidemia episode by definition instead of the current one.
  • Table 2: Is persistent candidemia lower for the non-survivors because they did not survive long enough to meet the definition of persistent candidemia? If so, may want to discuss or adjust analysis to account for that.
  • Figure 1: It looks like a lot of people did not survive long enough to have the opportunity to start treatment in the no treatment group. Perhaps it would help to clarify if day 0 on there represents day of specimen collection or day of diagnosis/results known.
  • L119: This references a supplementary table 1, but I do not see such a table.

Discussion

  • L215-217: I cannot find a definition of early v. delayed treatment in the authors’ paper. It would be helpful to clarify this for results interpretation.

General

  • Check italics for the various organisms described as well as spelling for organisms and drugs.

Author Response

Cortés, et al. describe a sub-investigation performed as part of a larger 7-country Latin American candidemia surveillance study. Specifically, they examine risk factors for mortality among people with candidemia in 4 hospitals. With increasing attention on candidemia in the past few years, it is valuable or would be of interest to readers to describe and share these mortality risk factors. While it’s well-written overall, the paper could be and, in some places, must be edited to correctly interpret the results. Here are some considerations for the authors:

  1. Abstract

L31: The catheter removal CI should be 0.00-0.49 according to table 3.

            Answer: The mistake was corrected in L31.

  1. Introduction

L38-39: For a paper about Colombia, it seems odd to have the opening sentence be about the United States. Is there perhaps a more general statement or a Colombia/Latin America/South America statement that could work there instead?

Answer: The reviewer is correct. A recent reference to Latin American epidemiology was added in L39.

  1. Methods
  • L75: Should this say “...current treatment for identified cancer”? Or is this intending to say current treatment for cancer overlapping with the time of having candidemia (more similar to the shock definition)?

Answer: The reviewer is correct, and it was corrected as suggested in L83.

  • Would it be possible to define the main outcome, mortality? It reads like this is describing all-cause mortality, but it would be helpful to clarify for the reader.

Answer: The following text was added: “Main outcome was defined as all-cause mortality during the hospital stay or in the following 28 days of the incident candidemia”.L119-L120.

  1. Results
  • L126-127: Could you put the number of ‘patients who underwent surgery’ in this sentence? It unclear currently whether you are saying 29 of the 69 patients in table 1 with surgery actually had >=2 surgeries or not and if the 12 abdominal procedures are included in the 34 of the table or not. Perhaps putting the number of surgeries patients had in the table would help?

Answer: The section was rephrased and expressed the total number of patients: “69 patients underwent surgery, and 29 of them had at least one other surgical procedure (12 of these second surgeries were abdominal procedures).”L140-142.

  • Table 1: Can the ages be added next to the age groups?

Answer: The age groups were added in the table.

  • Table 1: Can the total N be added to the table and clarify how the numbers are being presented? For instance, I initially assumed the table was out of 109 cases, but when you add up the numbers in each age group, it adds up to 153 yet the percentages for each age group appear to be out of 109 (e.g. 26/109 = 23.9%). Is it that there are 109 unique people involved but a substantial number moved up an age group between different candidemia episodes?

Answer: The reviewer is correct. There are 109 patients with unique candidemia episodes, and this was added to the table. The difference was made between 18 and 60 years (adults) and elderly (more than 60 years-old). The definition was corrected in the methods section (L82).

  • Table 1: What is the time period for central line, parenteral nutrition, and surgery?

Answer: We added the following text: Median time of catheter use was 12 days (IQR 7 – 19 days). The median time after the start of parenteral nutrition and the identification con candidemia was 17 days (IQR 10 – 24 days). We did not have information on dates of surgeries, so we could not calculate that information.  L160-162.

  • Table 1: There are a substantial number of things that seem like they would fit into table 1 that are mentioned in the text but are not presented in table 1 (e.g. APACHE score, mechanical ventilation, antifungals, etc.). I imagine readers may appreciate those being added to table 1.

Answer: The following information was added to table 1: mechanical ventilation, ICU admission, previous antibiotic use and previous antifungal use.

  • L153-154: The description of the susceptibility pattern as ‘excellent’ would be more appropriately placed in the discussion.

Answer: The phrase was removed from the results section.

  • L173: Is it correct that the upper end of the IQR is 70 days? Based on the 30-day candidemia episode criteria and the 30 day follow-up period, I would imagine that 70 days later would be counted towards a different candidemia episode by definition instead of the current one.

Answer: It was a typo, the correct number is 7.L195.

  • Table 2: Is persistent candidemia lower for the non-survivors because they did not survive long enough to meet the definition of persistent candidemia? If so, may want to discuss or adjust analysis to account for that.

Answer: The following phrase was added to the discussion: “Interestingly, non-survivors had a lower frequency of persistent candidemia, which might be due to the high mortality observed in the first days after the identification, specially in those without treatment.”L260-L262.

  • Figure 1: It looks like a lot of people did not survive long enough to have the opportunity to start treatment in the no treatment group. Perhaps it would help to clarify if day 0 on there represents day of specimen collection or day of diagnosis/results known.

Answer: In L191 we added to the existing phrase, so it stated now: “Treatment was started with a median delay of 2 days after the blood sample was taken”. And the following note was added to figure 1: “Time was counted beginning the day the blood sample was taken.”

  • L119: This references a supplementary table 1, but I do not see such a table.

Answer: The supplementary material was added with this letter.

  1. Discussion

L215-217: I cannot find a definition of early v. delayed treatment in the authors’ paper. It would be helpful to clarify this for results interpretation.

Answer: The following definition was added to the methods: “Delayed treatment was considered when the antifungal was started more  than 48 hours of the day the blood sample was taken.”L85-L86.

  1. General

Check italics for the various organisms described as well as spelling for organisms and drugs.

Answer: Misspelling and italics were corrected throughout the manuscript.

Reviewer 2 Report

The original article entitled: "Risk Factors for Mortality in Colombian Patients With Candidemia" by Cortes et al presents the results of laboratory-based surveillance of candidemia in 109 patients in 4 hospitals from Colombia.  The article focuses on the predisposing and risk factors for candidemia. 

Although the article does not add any additional information to current literarature to my opinion could be accepted after a major revision as reports data from Latin America.

Major comments:

  1. Abstract: authors report that the mortality rate was 35.7% and one of the risk factor for candidemia was the lack of antifungal treatment. Authors should clarify the different modalities for the approach of fungal infections used in their hospitals (prophylaxis for who and for how long, empirical treatment, ecc).
  2. In the material and methods section author declare to report data on risk factors for candidemia but to my opinion they should define the target population in the 4 hospitals (departments, children, neonates, surgery, cancer patients, patients under HSCT, patients with CVCs, diabetic patients, different underlying diseases).
  3. In the results authors should add more informations about differences between departments, units and ages groups (neonates, children, adults).
  4. Additionally, authors could provide informations about antifungal stewardship policies in the 4 hospitals and in Colombia in general.

Author Response

The original article entitled: "Risk Factors for Mortality in Colombian Patients With Candidemia" by Cortes et al presents the results of laboratory-based surveillance of candidemia in 109 patients in 4 hospitals from Colombia. The article focuses on the predisposing and risk factors for candidemia.

Although the article does not add any additional information to current literarature to my opinion could be accepted after a major revision as reports data from Latin America.

Major comments:

  1. Abstract: authors report that the mortality rate was 35.7% and one of the risk factor for candidemia was the lack of antifungal treatment. Authors should clarify the different modalities for the approach of fungal infections used in their hospitals (prophylaxis for who and for how long, empirical treatment, ecc).

Answer: The following was added to the methods: “There was no antifungal protocol in use in any the hospitals and no prophylaxis strategy for the prevention of such infections in any service of the institutions.”L68-L69.

  1. In the material and methods section author declare to report data on risk factors for candidemia but to my opinion they should define the target population in the 4 hospitals (departments, children, neonates, surgery, cancer patients, patients under HSCT, patients with CVCs, diabetic patients, different underlying diseases).

Answer: Thanks for the question. In the supplementary material the age group for each institution was available. Besides the gernal description fo the institutions, the following paragraph was added to the methods: “Patients from one hospital (FVL) came from a pediatric population with neonates or patients with less than one-year old. Patients from other Hospital (HUS) were adults. One general hospital was a referral center for the military health services in the country (HMC) and the other was a reference center for general population (HUSI).There was no antifungal protocol in use in any the hospitals and no prophylaxis strategy for the prevention of such infections in any service of the institutions.”.L64-L68.

  1. In the results authors should add more informations about differences between departments, units and ages groups (neonates, children, adults):

Answer: Some of this information is in the supplementary table, now available with this letter. Data on ICU admission was added to table 1, and the age groups were clarified also in table 1.

  1. Additionally, authors could provide informations about antifungal stewardship policies in the 4 hospitals and in Colombia in general.

Answer: There was no antifungal stewardship in any of the hospitals, and the following phrase was added in the discussion to account for this:” In Colombia, a previous study showed that a similar proportion of patients were treated with antifungal medications and had a similar mortality [12], showing that there is a need for prompt recognition and treatment initiation in patients with candidemia, and probably reflecting the need for stewardship programs on antifungals to provide appropriate treatments in those that require it and to avoid unnecessary use among the groups at risk.”. L251-L254

Round 2

Reviewer 2 Report

Dear all,

All corrections have been done and all queries have been answered.

The manuscript is suitable for publication.

Author Response

No thing to add. Thanks.